# Characterization of Carbapenem-Resistant *K. Pneumoniae* Isolated from Intensive Care Units of Zagazig University Hospitals

**DOI:** 10.3390/antibiotics11081108

**Published:** 2022-08-16

**Authors:** Nessma Hessin Mohamed Gandor, Ghada EL-Sayed Amr, Sahar Mohamed Saad Eldin Algammal, Alshymaa Abdullah Ahmed

**Affiliations:** 1Department of Clinical Pathology, Faculty of Medicine, Zagazig University, Alsharquia 44519, Egypt; 2Department of Anaesthesia and Surgical Intensive Care, Faculty of Medicine, Zagazig University, Alsharquia 44519, Egypt

**Keywords:** CRKP, *bla*
_NDM_, *bla*
_KPC_, *bla*
_OXA-48_, Egypt

## Abstract

The advent of carbapenem-resistant *Klebsiella pneumoniae* (CRKP) poses a significant challenge to public health, as carbapenems are typically employed as a last resort to treat nosocomial infections caused by such organisms, especially in intensive care units (ICUs). This study aims to characterize the CRKP isolated from patients admitted to the Zagazig University Hospitals (ZUHs) ICU in Egypt. About 56.2%, 41.0%, and 32.4% of the isolates indicated the presence of *bla*_NDM_, *bla*_OXA-48_, and *bla*_KPC_, respectively. Carbapenemase-encoding genes were found in many isolates, and *bla*_NDM_ was the most predominant gene. Nevertheless, this situation has become a heavy burden in developing countries, including Egypt, and is associated with substantial morbidity, mortality, and increased healthcare expenses.

## 1. Introduction

*Klebsiella pneumoniae*, a gram-negative, non-motile, opportunistic pathogen, is one of the leading causes of hospital-acquired infections such as urinary tract infections, pneumonia, septicemia, and meningitis [1]. Infections from *K. pneumoniae* are usually associated with significant morbidity and mortality, especially among immunocompromised patients [2]. It is one of the most frequent pathogens that exhibit resistance to multiple antibiotics globally [3]. Through plasmids and transposons, these bacteria can effortlessly acquire and transfer genetic resistance determinants [4]. The acquisition of these genes leads to the production of β-lactamases that can specifically hydrolyze the β-lactam ring, leading to an inactivated product unable to inhibit the bacterial transpeptidase anymore [5]. Extended-spectrum β-lactamases (ESBLs) are the most common β-lactamases [4]. ESBLs can hydrolyze extended-spectrum penicillins, cephalosporins, and monobactams, limiting therapeutic options to carbapenems [6]. As a result, carbapenems are frequently employed as a last option in treating infectious diseases triggered by multidrug-resistant bacteria [7]. Nevertheless, selection pressure due to carbapenem overuse or misuse has led to the creation of carbapenem-resistant *Enterobacteriaceae* (CRE). CRKP is the most prevalent isolate among the increasing numbers of documented CRE cases worldwide [8].

These usually involve the synthesis of different classes of carbapenemases, hyperproduction of AmpC β-lactamases with an outer membrane porin mutation, and production of ESBL with either a porin mutation or drug efflux mechanisms for carbapenem resistance in CRKP. Carbapenemase production is the most described mechanism of carbapenem resistance in CRKP [9]. All β-lactam antibiotics, particularly carbapenems, monobactams, and extended-spectrum cephalosporins, can be hydrolyzed by carbapenemases [10]. 

Carbapenemases can be classified into three functional classes amongst the four classes of β-lactamases defined by the Ambler classification; (i) class A contains serine in the active site of the enzyme such as *K. pneumoniae* carbapenemase (KPC) [11], (ii) metallo-β-lactamases (MBLs) of class B, including Verona integron metallo-β-lactamase (VIM), imipenemase metallo-β-lactamase (IMP), as well as New Delhi metallo-β-lactamase (NDM) [12], and (iii) oxacillin-hydrolyzing β-lactamases (OXA) of class D [13]. The carbapenem hydrolyzing genes are frequently encoded on mobile genetic elements alongside other antibiotic resistance genes [12]. For example, genes encoding for aminoglycoside resistance (aminoglycoside-modifying enzymes “AMEs” and 16S ribosomal RNA methyltransferases “16S-RMTases”) were found to be prevalent in CRKP [14]. Therefore, the widespread of carbapenemase-encoding genes ultimately resulted in antimicrobial resistance gene co-transfer and increased the prevalence of bacterial infections [12].

Testing for carbapenemase-producing *K. pneumoniae* strains has become increasingly important owing to the dramatic rise in carbapenem resistance among *K. pneumoniae* isolates [15]. Carbapenem resistance has increased in Egypt, and multiple investigations found that carbapenem resistance was present in 44.3% of *K. pneumoniae* samples [16]. However, carbapenem resistance in *K. pneumoniae* from the Delta region has been a research topic and few studies are available. Because carbapenems are commonly prescribed as empirical treatment in ICUs at Zagazig University Hospitals (ZUHs), Egypt, this study aims to determine carbapenem phenotypic and genotypic resistance characteristics in CRKP clinical isolates obtained from patients in ICUs.

## 2. Results

### 2.1. Identification of the Bacterial Isolates

Out of 815 pathogens isolated from ICUs patients, 560 (68.7%) were gram-negative bacilli, 214 (26.3%) were gram-positive cocci, and 41 (5.0%) were *Candida spp.* Of 560 gram-negative bacilli isolates, 180 (32.1%) were identified as *K. pneumoniae*. 

### 2.2. Characteristics of K. pneumoniae Isolates

This study recruited 180 individuals with confirmed *K. pneumoniae* infections; 119 had CRKP infections. CRKP isolates were collected from 61 female and 58 male patients. Patients’ ages ranged from 2 days to 98 years, with a median of 44 years. The median length of stay in the ICU was 14 days (5–30). Most of the isolates (91.6%) were hospital-acquired infections. The isolates were recovered in 39.5% of respiratory samples, 24.4% of urine tests, 18.5% of blood samples, 8.4% of pus samples, 4.2% of central venous catheter (CVP) samples, and 2.5% of both cerebrospinal fluid (CSF) and peritoneal fluid samples (Table 1).

### 2.3. Antimicrobial Susceptibility Testing

CRKP isolates were 100% resistant to extended-spectrum penicillin (piperacillin, piperacillin/tazobactam, and ticarcillin), third generation cephalosporins (ceftazidime), and fourth generation cephalosporins (cefepime), macrolides (azithromycin), and carbapenems (imipenem, and meropenem). Additionally, they were highly resistant to aminoglycosides (tobramycin 95.0%, gentamicin 83.2%, and amikacin 88.2%), quinolones (pefloxacin 95.0%, and ciprofloxacin 98.3%), semi-synthetic tetracyclines (minocycline 89.9%), as well as sulfamethoxazole/trimethoprim 78.2%. Moreover, 36.1% of isolates were resistant to glycecyclines (tigecycline), whereas 10.9% were resistant to polymixins (colistin). The most sensitive antibiotics were colistin, 89.1%, and tigecycline, 55.5% (Table 2).

### 2.4. Confirmation of Carbapenem Resistance

According to the MEM E-test results, 110/180 (61.1%) of the isolates were meropenem non-susceptible; the range of MIC values was 0.002–32 µg/mL.

VITEK2 compact susceptibility findings and MEM E-test results are shown in Table 3 and Figure 1. There was a moderate significant agreement in the results, with the Kappa level for the E-test = 0.605 and a *p*-value of <0.0001.

### 2.5. Phenotypic Carbapenemase Detection

Carbapenemase activity was detected in 58/119 (48.7%) by the MHT method, 106/119 (89.1%) by the mCIM method. 

### 2.6. Detection of Carbapenemase-Encoding Genes

The isolates were examined by multiplex PCR to detect *bla*_NDM_, *bla*_IMP_, *bla*_VIM_, *bla*_KPC_, and *bla*_OXA-48_. A total of 105 (88.2%) isolates harbored one or more of the evaluated carbapenemase-encoding genes (Table 4). The frequency of the evaluated genes was: *bla*_NDM_ (56.2%), *bla*_OXA-48_ (41.0%), and *bla*_KPC_ (32.4%). *bla*_NDM_ was the most predominant gene. Neither *bla*_IMP_ nor *bla*_VIM_ was detected in any of the isolates.

In 22/105 (21.0%) of the isolates, carbapenemase-encoding genes were co-harbored. In all, 9/105 of the isolates (8.6%) presented all three genes, while 4/105 of the isolates (3.8%) presented both *bla*_KPC_ and *bla*_NDM_ genes, 3/105 of the isolates (2.9%) presented both *bla*_KPC_ and *bla*_OXA-48_ genes, and 6/105 of the isolates (5.7%) presented both *bla*_NDM_ and *bla*_OXA-48_ genes (Table 5 and Figure 2).

### 2.7. Demographic Characteristics of CRKP Patients According to the Detected Carbapenemase-Encoding Genes

Females showed a higher distribution of *bla*_KPC_ and *bla*_OXA-48_, but this was statistically insignificant (Table 6).

### 2.8. Correlation of the Phenotype and Genotype of Carbapenem Resistance

Among the 58 MHT-positive isolates, 18 were positive for *bla*_KPC_, 19 for *bla*_OXA-48_, and 15 presented two or even more carbapenemase-encoding genes concurrently. Only two isolates harboring the *bla*_NDM_ gene were MHT positive, but none of these genes were present in the remaining four MHT positive isolates.

Out of 106 mCIM-positive isolates, 18 were positive for *bla*_KPC_, 37 for *bla*_NDM_, 25 for *bla*_OXA-48_, 22 for two or more carbapenemase-encoding genes, and 4 for none (Table 7).

### 2.9. Susceptibility Testing to New Therapeutic Agent (CZA)

Regarding susceptibility to the new therapeutic agent, ceftazidime/avibactam (CZA), only 28/119 (23.5%) of CRKP isolates were susceptible to it. All isolates harboring *bla*_NDM_ were resistant to it (100% resistance), while the resistance rate in serine-producing isolates was 56.5% (Table 8). 

## 3. Discussion

In recent years, CRKP has spread to several countries across the globe [17]; Egypt is not an exception. Healthcare systems and the public are at risk because of the high occurrence of CRKP isolates throughout the Mediterranean, especially in Egypt [18]. In earlier studies [16], CRKP isolates were shown to be prevalent in Egypt at a rate of 44.3% [16], especially among cancer patients [19] and ICU admitted patients [2]. Our study provides further evidence of the high prevalence of CRKP among ICU admitted patients. Hence, the characterization of carbapenem-resistant isolates is the first step on the road map for controlling the spread of these isolates [20]. Consequently, this investigation examined our institution’s phenotypical and genotypical features of CRKP isolates.

Prior research indicated that ICU admission was a substantial risk factor for developing CRKP [21]. The participants in this study were individuals who had been admitted to the ICU.

CRKP occurrence was relatively higher among patients aged 20 to 60 years, but this was not statistically significant, especially in comparison to other age groups. In all, 18.4%, 32.7%, and 48.7% of ICU patients stayed in the hospital for ≤7 days, (8–14) days, and ˃14 days, respectively. Moreover, the incidence rate of CRKP was elevated by lengthier hospital stays, but different durations were statistically insignificant. According to a study in Egypt, a prolonged stay in the ICU before specimen collection constituted a significant risk factor for carbapenem resistance [22]. In another study in Egypt, Kairy et al. found that 37.5% of their patients had a hospital admission of more than ten days, accompanied by 28.2% who spent eight to ten days in the hospital [23].

Sputum samples were the most common source of CRKP in this study. Similarly, research in Indonesia [24] and Egypt [2] found that sputum included the highest number of bacteria with carbapenemase-encoding genes. Due to cross-infection with multidrug-resistant clones or long-term exposure to antibiotics, resistance determinants accumulate in respiratory tract microbiota, increasing antibiotic resistance. Resistant microbes may lead to respiratory tract infections in the future [25].

An MDR profile was found in the antimicrobial susceptibility testing of CRKP isolates, with 100% of the isolates being resistant to piperacillin, piperacillin/tazobactam, cefepime, ceftazidime, azithromycin, and ticarcillin. In comparison, 95.0% of the isolates were highly resistant to tobramycin, gentamicin (83.2%), pefloxacin (95%), and ciprofloxacin (98.3%). According to our findings, cefepime (86.5%), piperacillin/tazobactam (72.5%), ciprofloxacin (71.5%), tobramycin (52.0%), and gentamicin (45.0%) were more susceptible than any of those reported by Zafer et al. [19].

Colistin and tigecycline are the last choices for treating CRKP infections [26]. Unfortunately, 10.9% and 36.1% of our CRKP isolates were resistant to colistin and tigecycline, respectively. Many studies have attributed resistance to colistin and tigecycline to the spread of mobile plasmid-mediated colistin resistance determinants such as *mcr* genes [27] and mobile tigecycline resistance determinants such as flavin-dependent mono-oxygenase *tet* (X) and *tmex*CD1*-topJ1* variants [28]. The low resistance to colistin (10.9%) in our study compared to other antibiotics could be explained by the nephrotoxicity and neurotoxicity side effects of colistin limiting their prescription, especially in severely ill patients such as our study group [29]. 

An E-test is a conventional approach; however, it can take a long time to administer. In this research, the findings of the E-test and VITEK 2 were correlated to a moderate degree, so we recommended using VITEK 2 results instead of waiting for the E-test’s findings to arrive.

The phenotypic identification of carbapenemases was carried out using MHT and mCIM before multiplex PCR was used to determine the carbapenemase-encoding genes. MHT was negative for 61 out of the 119 tested isolates. However, upon screening of carbapenemase-encoding genes by multiplex PCR, 51 were found to harbor carbapenemase-encoding genes. Miriagou et al. [30] reported that the existence of metallo-β-lactamase producing isolates with limited carbapenemase activity might explain the false negatives of MHT.

On the other hand, false-positive results were frequently reported with MHT, particularly in isolates that produce high levels of AmpC β-lactamases (cephalosporinases) or CTX-M-type ESBLs [11]. Our finding provides further evidence of the low sensitivity and specificity of MHT, especially for *bla*_NDM_ producers, as described in previous studies [31,32,33]. Although MHT has been widely used in the clinical laboratory for carbapenemase detection, it cannot identify the class of carbapenemase-encoding genes involved. Identifying the class of the carbapenemase gene in CRKP is essential for therapeutic decision-making. AST results alone may be sufficient for selecting antimicrobial therapy, but different antibiotics may exhibit varying levels of susceptibility to the action of carbapenemases [34]. 

Some carbapenemases can hydrolyze carbapenems very efficiently, while others may have a lesser extent. Some carbapenemases are also active against broad-spectrum cephalosporins, while others are not [35]. For example, class B metallo-β-lactamases (*bla*_NDM_, *bla*_IMP_, and *bla*_VIM_) inactivate the most available β-lactams [8]. Class A carbapenemase-encoding gene (*bla*_KPC_) can hydrolyze nearly all β-lactams, including carbapenem, penicillin, aztreonam, and cephalosporin. However, it is inhibited by β-lactamase inhibitors as avibactam. Avibactam is also effective against ESBLs, Ampc β-lactamases, and *bla*_OXA-48_ [36].

According to our findings, carbapenemase activity in CRKP isolates was better detected by mCIM than by MHT. Research in the United States by Pierce et al. found that 91 of the 92 carbapenemase-infected isolates tested positive for mCIM [37]. The mCIM test is uncomplicated, affordable, requires no special equipment, and is simple to interpret. Nevertheless, the lengthy incubation period (from eight hours to overnight) cannot be neglected. Furthermore, the carbapenemase class could not be verified by mCIM as another limitation.

Nevertheless, the presence of carbapenemase-encoding genes does not always cause resistance to carbapenems. On the other hand, resistance in a non-carbapenemase-producing strain has been linked to secondary processes, including decreased outer membrane permeability, increased efflux pumps, or hyperexpression of broad-spectrum cephalosporins [38]. Therefore, a better patient outcome could be ensured by making treatment decisions based on genotypic and phenotypic factors.

Carbapenemase-encoding genes can be identified using known primers and molecular detection by PCR. Using the multiplex PCR method, the 119 CRKP isolates involved in this investigation were evaluated for the presence of the most frequently documented carbapenemase-encoding genes in *K. pneumoniae*. Carbapenemase-encoding genes were found on 88.2% of the samples in the current investigation, compared to 44.3% in another research by El-Sweify et al. [16].

According to our research, *bla*_NDM_ (56.2%) was the most common carbapenemase-encoding gene found in the investigated CRKP isolates, followed by the class D carbapenemase *bla*_OXA-48_ (41.0%), and then the *bla*_KPC_ gene (32.4%). These findings were in contrast to previous studies in which class D carbapenemases (*bla*_OXA-48_ = 58.0%) were the most common in *K. pneumoniae* [39,40]. Since they are encoded on a range of important mobile conjugative plasmids, the predominance of *bla*_NDM_ might be explained by the fact that they are encoded on a range of highly mobile conjugative plasmids, which enable horizontal inter- as well as intra-transfer rather than clonal spread between bacteria [41].

Since plasmids encoding for *bla*_NDM_ commonly incorporate resistance genes to almost all common antibiotics, the dominance of *bla*_NDM_-harboring bacteria must be considered [42]. 

Our current study revealed that carbapenemase-encoding genes were co-harbored in 22/105 (21.0%), which was lower than earlier findings by Emira et al. and El-Domany et al., who found that isolates harboring multiple carbapenemase-encoding genes were as high as 48% and 57.9%, respectively [15,43]. One of our remarkable findings was the uncommon combination of the three carbapenemase-encoding genes (*bla*_KPC_, *bla*_NDM_, and *bla*_OXA-48_), which had never been identified among isolates from Egypt.

Many gene cassettes were found in *bla*_NDM_-infected bacteria encoding different carbapenemases, which might explain why many carbapenemase-encoding genes were co-present [44]. A single carbapenemase isolate that contains many carbapenemases is very resistant to therapy because it expands its overall hydrolytic range [45].

Even though AMR is a global issue, the effect on developing economies is disproportionately high [46,47,48]. This significant burden on developing countries could be the direct outcome of inadequate strategies for novel antibiotics, increased economic burden, and limited capacity to provide second-line antibiotics, which may be more expensive and have worse consequences.

Regarding susceptibility to the new therapeutic agent (CZA), the resistance rate was 76.5%. This is a very high resistance rate compared to the results of the Surveillance of Multicenter Antimicrobial Resistance in Taiwan (SMART) in 2017, which reported 100% susceptibility rate of CRKP isolates to CZA [49]. All isolates harboring *bla*_NDM_ either alone or with other carbapenemase-encoding genes were resistant to CZA (100% resistance), while the resistance rate in serine-producing isolates was 56.5%. Our results partially agree with previous studies, which mentioned that most MBL-positive isolates were resistant to CZA with a resistance rate ranging from 90.8% to 100% [50]. Avibactam action involves the formation of non-covalent linkage to a sensitive β-lactamase binding site, followed by the covalent acylation of a β-lactamase binding site at serine residue [50]. Therefore, it is not effective against MBL-producing isolates. However, it is worth mentioning that avibactam, by its strong activity against class A β-lactamases and AmpC lactamases, can restore the activity of aztreonam against MBL-producing isolates [51]. Emeraud et al. have stated that the combination of aztrenam (ATM)- ceftazidime (CZ)- avibactam (AVI) has a treating activity against 86% of MBL-producing isolates [36].

The presence of multiple β-lactamases, chemical modification of the target site, drug efflux mechanisms and changes in cell permeability could explain the resistance found in serine-producing isolates [51]. 

Nevertheless, poor infection prevention and control (IPC) efforts in developing countries have increased AMR prevalence [52]. In contrast, poor hospital-based antibiotic use regulation and excessive use of antibiotics in food-producing animals are the main contributing factors responsible for increased AMR prevalence in developed countries [53].

## 4. Materials and Methods

### 4.1. Study Design

Between December 2019 and April 2021, this cross-sectional observational study was conducted in the Clinical Pathology Department of Zagazig University Hospitals (ZUHs) in Alsharqiya governorate, Egypt. This research included patients diagnosed with *K. pneumoniae* infections from various ICUs of ZUHs, a set of tertiary referral hospitals that serve five governorates in eastern Egypt. These ICUs include anesthesia ICU (30-bed unit), chest ICU (15-bed unit), internal medicine ICU (54-bed unit), neurosurgery ICU (20-bed unit), oncology ICU (10-bed unit), pediatric ICU (14-bed unit), stroke ICU (9-bed unit), surgical ICU (24-bed unit), and tropical ICU (24-bed unit). 

### 4.2. Ethical Approval

The current study was approved by the Institutional Review Board (IRB) committee of Zagazig University (no. ZU-IRB # 5215/5-3-2019).

### 4.3. Case Diagnosis

In the presence of clinical symptoms and evidence of infection, a patient was diagnosed with *K. pneumoniae* if isolated from a sterile site such as blood, peritoneal fluid, or cerebral spinal fluid. Coughing, dyspnea, and fever were all symptoms of pneumonia, as well as the development of an infiltrate on chest radiography and the presence of more than 10^4^ colony-forming units/mL of purulent tracheal secretions or bronchoalveolar lavage fluid. Leukocytes and microorganisms were detected and quantified using gram staining. Bacterial growth of more than 15 colony-forming units in roll-plate culture or 10^3^ colony-forming units in quantitative sonication was used to identify a catheter tip infection. There must be at least 10,000 microorganisms/mL isolated to diagnose a urinary tract infection, as well as at least two of the signs mentioned above and symptoms: frequency, dysuria, or pyuria (>10 white blood cells/HPF) [2].

### 4.4. Data Collection and Bacterial Strains

In sterile containers, samples of urine, sputum, blood, pus, cerebrospinal fluid (CSF), CVP, and peritoneal fluid were gathered. When *K. pneumoniae* was found in more than one specimen from the same patient, the researchers only considered the first sample.

### 4.5. Microbiological Identification

Blood agar and MacConkey agar were used to cultivate clinical specimens (Oxoid Co., Altrincham, UK). Standard microbiological procedures (colony morphology and gram stain) and MALDI-TOF (VITEK^®^ MS) were used to identify isolates.

### 4.6. Antimicrobial Susceptibility Testing

The VITEK^®^ 2 compact system and AST-GN 222 card (BioMérieux, Marcy L’Etoile, France) were used to test antimicrobial susceptibility. The disk diffusion method (15 µg) was used to test susceptibility to tigecycline. The findings were interpreted following the CLSI 2019 guidelines [54]. Due to the lack of established CLSI breakpoints for TGC at present, the Food and Drug Administration (FDA) breakpoints issued for Enterobacteriaceae (≥19 mm, susceptible; 15–18 mm, intermediate; and ≤14 mm, resistant) (https://www.accessdata.fda.gov/drugsatfda_docs/label/2013/021821s026s031lbl.pdf, accessed on 30 January 2021) were used for the interpretation of the results [55]. The obtained isolates were kept in 50% glycerol at −80 °C for further examination.

### 4.7. Confirmation of Carbapenem Resistance

All isolates were retested using meropenem E-test strips (MRP) with a concentration gradient of 0.002–32 µg/mL to ensure a precise MIC measurement (Liofilchem, Roseto degli Abruzzi, Italy). The CLSI 2019 breakpoints were used to interpret the findings. A 4 µg/mL cut-off value was used to define resistance, while a 1 µg/mL cut-off determined susceptibility [54].

### 4.8. Phenotypic Carbapenemase Detection

The following phenotypic tests were used for screening carbapenemase production:

#### 4.8.1. Modified Hodge Test (MHT)

Muller–Hinton agar plates were streaked with a 1:10 dilution of 0.5 McFarland standard *E. coli* strain ATCC 25922 suspension, and 0.5–4.5 mL of saline (45%) were added. Then, it was placed in the middle of the plate and supplemented with meropenem (10 µg). The test isolate was smeared straight from the disk to the plate’s edge. The plates were kept overnight at 35 to 37 °C. CLSI guidelines were followed to interpret positive and negative results [56].

#### 4.8.2. Modified Carbapenem Inactivation Method (mCIM)

For 15 s, 1 µL of calibrated loopful of organism suspension was vortexed in 400 µL of water. The suspension was then aseptically supplemented with a 10 µg meropenem disk. The suspension, including the disk, was incubated at 35–37 °C for 4 h. The meropenem disk was then removed from the suspension using a 10 µL inoculating loop; the loop was dragged along the edge of the eppendorf to eliminate extra liquid and then placed on a Muller–Hinton agar plate inoculated with a 0.5 McFarland suspension of standard carbapenem-resistant *E. coli* strain ATCC 25922. Plates were also incubated overnight at 35–37 °C; the inhibition zone surrounding the disk was determined. Carbapenemase positivity was defined as an inhibition zone of 6–15 mm or colonies inside a 16–18 mm zone. On the other hand, a carbapenem inhibition zone more than 19 mm was considered negative [37].

### 4.9. Molecular Detection of Carbapenemase-Encoding Genes

DNA extraction was conducted according to the manufacturer’s guidelines by employing the G-spin^TM^ Genomic DNA Extraction Kit (iNtRON Biotechnology, Inc., Gyeonggi-do, Korea). Multiplex PCR was used to detect the five significant carbapenemase-encoding genes in *K. pneumoniae* (*bla*_KPC_, *bla*_NDM_, *bla*_VIM,_
*bla*_IMP_, and *bla*_OXA-48_) [12]. A total volume of 30 µL was adequately prepared for the PCR reaction mixture, including 5 µL (100 ng) of template DNA, 15 µL of PCR master mix, and 2 µL (5 pmol) of each primer (Table 9) [57,58]. The volume was then completed with nuclease-free water up to 30 µL. For the amplification, the following thermal cycling conditions were employed: a five-minute initial denaturation step at 95 °C, followed by 15 cycles of DNA denaturation at 95 °C for 30 s, primer annealing at 58 °C for 40 s, and primer extension at 72 °C for one minute. This was accompanied by 25 cycles of DNA denaturation at 95 °C for 30 s, primer annealing at 50 °C for 40 s, and primer extension at 72 °C for one minute. Then, gel electrophoresis and UV visualization were conducted for the PCR products. 

### 4.10. Susceptibility Testing to New Therapeutic Agent (CZA)

The new therapeutic agent, ceftazidime/avibactam (CZA) (30/20 µg) obtained from (Oxoid Co., Altrincham, UK), was tested against CRKP isolates using the disk diffusion method. The diameters of the inhibition zones were interpretated according to CLSI 2019 guidelines [54].

## 5. Conclusions

According to the current findings, CRKP was responsible for a significant number of HAI cases in the ICUs of ZUHs. Antibiotic resistance was shown to be widespread in our study. A significant number of the isolates had carbapenem-resistance genes, with *bla*_NDM_ being the most common. The co-presence of multiple carbapenemase-encoding genes was found in many CRKP isolates. Evidence-based IPC methods and antibiotic stewardship programs must be implemented immediately to avoid the spread of the CRKP. New antimicrobial agents against CRKP, such as aztreonam-ceftazidime-avibactam should be tested and included in treating these resistant strains. Early recognition of carbapenem-resistant isolates is critical in restricting transmission; however, it is an epidemiologic and economic issue, particularly in developing countries.

## 6. The Limitations of This Study

The absence of multiple gene types is one of the study’s limitations. Moreover, due to a lack of resources, we have not undertaken whole-genome sequencing (WGS).

## 7. Recommendations

Multi-locus sequence technique (MLST), pulsed-field gel electrophoresis (PFGE), whole-genome sequencing (WGS), and plasmid analysis are recommended to be performed in future studies in conjunction with phenotypic tests and PCR.

Susceptibility testing for newer agents such as ceftolozane/tazobactam, cefiderocol meropenem/vaborbactam, and imipenem/relebactam is recommended. 

## Figures and Tables

**Figure 1 antibiotics-11-01108-f001:**
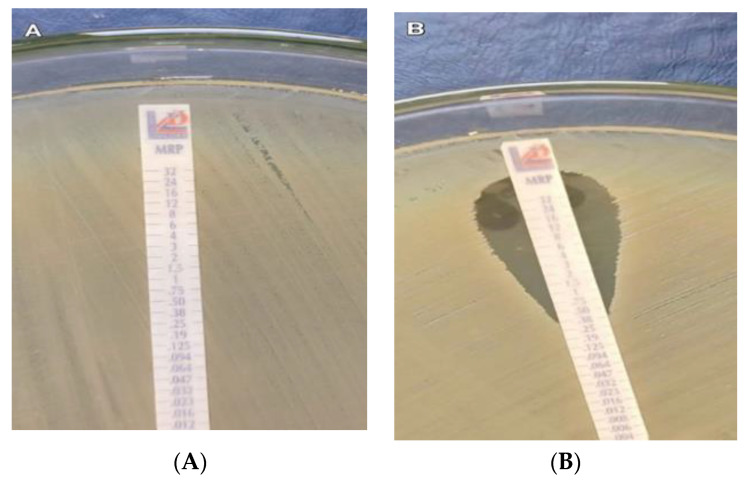
Meropenem E-test; (**A**) Muller–Hinton agar shows carbapenem-resistant *K. pneumoniae* with MIC ≥ 32 µg/mL, and (**B**) Muller–Hinton agar shows carbapenem-sensitive *K. pneumoniae* with MIC < 0.38 µg/mL.

**Figure 2 antibiotics-11-01108-f002:**
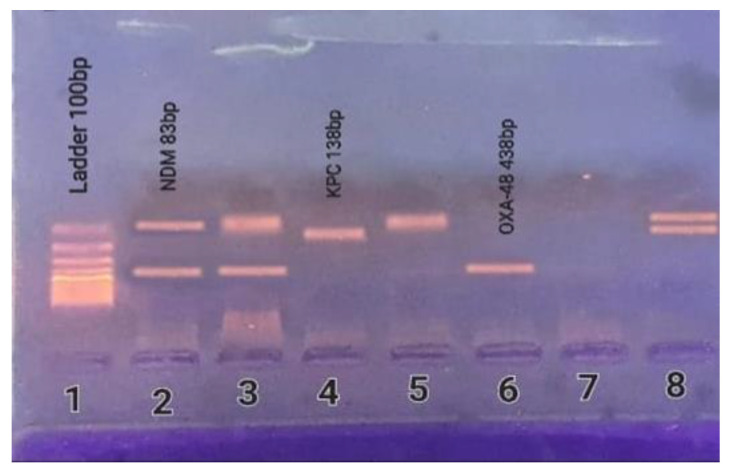
PCR results for carbapenemase-encoding genes; lane (**1**): DNA ladder 100 bp, lanes (**2,3**): positive for *bla*_NDM_ (83 bp) and *bla*_OXA-48_ (438 bp), lane (**4**): positive for *bla*_KPC_ (138 bp), lane (**5**): positive for *bla*_NDM_, lane (**6**): positive for *bla*_OXA-48_, lane (**7**): negative, and lane (**8**): positive for *bla*_NDM_ and *bla*_KPC_.

**Table 1 antibiotics-11-01108-t001:** Demographic data of CRKP patients.

	*n*	%
**Gender**		
Female	61	51.3%
Male	58	48.7%
**Age (year):**		
Mean ± SD	43.98 ± 22.94	
Range	44 years (2 days–98 years)	
**Age groups:**		
Infant (<2 years)	4	3.4%
Children&Adolescents	15	12.6%
Adult	66	55.5%
Old Age	34	28.6%
**Department (ICU):**		
Anesthesia	44	37%
Chest	4	3.4%
Internal Medicine	32	26.9%
Neurosurgery	8	6.7%
Oncology	2	1.7%
Pediatric	10	8.4%
Stroke	8	6.7%
Surgical	9	7.6%
Tropical	2	1.7%
**Type of Sample:**		
Blood	22	18.5%
CSF	3	2.5%
CVP	5	4.2%
Peritoneal Fluid	3	2.5%
Pus	10	8.4%
Sputum	47	39.5%
Urine	29	24.4%

**Table 2 antibiotics-11-01108-t002:** Distribution of the studied isolates according to antimicrobial susceptibility.

	Sensitive	Intermediate	Resistant
	*n* (%)	*n* (%)	*n* (%)
**Piperacillin/Tazobactam**	0 (0.0)	0 (0.0)	119 (100)
**Piperacillin**	0 (0.0)	0 (0.0)	119 (100)
**Tobramycin**	6 (5.0)	0 (0.0)	113 (95.0)
**Cefepime**	0 (0.0)	0 (0.0)	119 (100)
**Imipenem**	0 (0.0)	0 (0.0)	119 (100)
**Gentamicin**	15 (12.6)	5 (4.2)	99 (83.2)
**Ceftazidime**	0 (0.0)	0 (0.0)	119 (100)
**Meropenem**	0 (0.0)	0 (0.0)	119 (100)
**Amikacin**	14 (11.8)	0 (0.0)	105 (88.2)
**Sulfamethoxazole-Trimethoprim**	26 (21.8)	0 (0.0)	93 (78.2)
**Azithromycin**	0 (0.0)	0 (0.0)	119 (100)
**Minocycline**	7 (5.9)	5 (4.2)	107 (89.9)
**Ticarcillin**	0 (0.0)	0 (0.0)	119 (100)
**Pefloxacin**	0 (0.0)	6 (5.0)	113 (95.0)
**Ciprofloxacin**	0 (0.0)	2 (1.7)	117 (98.3)
**Tigecycline**	66 (55.5)	10 (8.4)	43 (36.1)
**Colistin**	106 (89.1)	0 (0.0)	13 (10.9)

**Table 3 antibiotics-11-01108-t003:** Correlation between VITEK2 compact and MEM E-test.

	VITEK2 Compact		Kappa Agreement	*p*-Value
	Susceptible (*n* = 61)	Resistant (*n* = 119)		
**MEM E-test**				
Susceptible (*n* = 70)	49 (80.3%)	21 (17.6%)	0.605	<0.0001
Resistant (*n* = 110)	12 (19.7%)	98 (82.4%)		

**Table 4 antibiotics-11-01108-t004:** Frequency of carbapenemase-encoding genes presence among CRKP isolates.

CRKP Isolates	No = 119	
	No	%
Carbapenmase Gene Presence	105	88.2%
No Gene Presence	14	11.8%

**Table 5 antibiotics-11-01108-t005:** Genetic profile of CRKP isolates.

	No = 105	
Carbapenemase-Encoding Gene	No	%
*bla* _NDM_	40	38.1%
*bla* _OXA-48_	25	23.8%
*bla* _KPC_	18	17.1%
*bla*_NDM_ + *bla*_OXA-48_ + *bla*_KPC_	9	8.6%
*bla*_NDM_ + *bla*_KPC_	4	3.8%
*bla*_OXA-48_ + *bla*_KPC_	3	2.9%
*bla*_NDM_ + *bla*_OXA-48_	6	5.7%

**Table 6 antibiotics-11-01108-t006:** Correlation between demographic characteristics of CRKP patients and the detected carbapenemase-encoding genes.

	KPC		NDM		OXA-48	
	Positive *n* = 34	Negative *n* = 85	Positive *n* = 50	Negative *n* = 69	Positive *n* = 43	Negative *n* = 76
**Age**						
**Mean ± (SD)**	**42.4 ± (19)**	**44.4 ± (23.8)**	**41.8 ± (24.1)**	**45.3 ± (21.2)**	**46.6 ± (22.4)**	**42.3 ± (22.5)**
Mann–Whitney	−0.5		−0.75		−0.92	
*p*-value	0.62		0.46		0.36	
**Sex**						
Male	13	45	28	30	18	40
Female	21	40	22	39	25	36
Chi-square (χ^2^)	2.1		1.82		1.28	
*p*-value	0.15		0.18		0.26	

**Table 7 antibiotics-11-01108-t007:** Agreement between genotypic and phenotypic tests.

	PCR Genes		Kappa Agreement	*p*-Value
	Negative (Susceptible) *n* = 14 (%)	Positive (Resistant) *n* = 105 (%)		
**MHT**				
Susceptible (*n* = 61)	10 (71.4)	51 (48.6)	0.093	0.108006
Resistant (*n* = 58)	4 (28.6)	54 (51.4)		
**mCIM**				
Susceptible (*n* = 13)	10 (71.4)	3 (2.6)	0.328	<0.00001
Resistant (*n* = 106)	4 (28.6)	102 (97.4)		

**Table 8 antibiotics-11-01108-t008:** Susceptibility of CRKP isolates to CZA.

CZA			
	Sensitive No (%)	Intermediate No (%)	Resistanat No (%)
CRKP (119)	28 (23.5)	-	91 (76.5)
-Carbapenemase-encoding genes negative (14)	8 (57.1)	-	6 (42.9)
-Carbapenemase-encoding genes positive (105)	20 (19.0)	-	85 (81)
● *bla*_NDM_ positive (40)	0 (0.0)	-	40 (100)
● *bla*_OXA-48_ positive (25)	11 (44.0)	-	14 (56.0)
● *bla*_KPC_ positive (18)	8 (44.4)	-	10 (55.6)
● *bla*_NDM_ + *bla*_OXA-48_ + *bla*_KPC_ positive (9)	0 (0.0)	-	9 (100)
● *bla*_NDM_ + *bla*_OXA-48_ positive (6)	0 (0.0)	-	6 (100)
● *bla*_NDM_ + *bla*_KPC_ positive (4)	0 (0.0%)	-	4 (100%)
● *bla*_OXA-48_ + *bla*_KPC_ positive (3)	1 (33.3%)	-	2 (66.7%)

**Table 9 antibiotics-11-01108-t009:** Sequence and expected product sizes of primers used to amplify *bla*OXA-48, NDM, KPC, IMP, and VIM genes.

Amplicon Size (bp)	Nucleotide Sequence (5′-3′)	Primers
438	F-(GCGTGGTTAAGGATGAACAC)	*bla* _OXA-48_
	R-(CATCAAGTTCAACCCAACCG)	
83	F-(CATTAGCCGCTGCATTGATG)	*bla* _NDM_
	R-(GTCGCCAGTTTCCATTTGCT)	
138	F-(TGCAGAGCCCAGTGTCAGTTT)	*bla* _KPC_
	R-(CGCTCTATCGGCGATACCA)	
740	F-(TGAGCAAGTTATCTGTATTC)	*bla* _IMP_
	R-(TTAGTTGCTTGGTTTTGATG)	
747	F-(TCTACATGACCGCGTCTGTC)	*bla* _VIM_
	R-(TGTGCTTTGACAACGTTCGC)	

## Data Availability

All data are available on request from authors.

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
