# Peer review of "Characterization of Carbapenem-Resistant K. Pneumoniae Isolated from Intensive Care Units of Zagazig University Hospitals"

_antibiotics, 2022, doi:10.3390/antibiotics11081108_

Round 1

Reviewer 1 Report

Nessma Hessin Mohamed Gandor et al. characterized carbapenem-resistant isolates of Klebsiella pneumoniae from intensive care units. In this single-center study authors use several phenotypic tests, as well as a multiplex PCR to characterize CRKP. Although the results are concerning, the study is lacking novelty. The biggest shortcoming is probably the fact that it is a single-center study with a very limited methodology: Authors use a multiplex PCR which is capable to detect only "the big five" carbapenemase-encoding genes.

There are several other shortcomings. To name a few:

- The manuscript is written awkwardly

- There is some mixup of different concepts, e.g., a) infections do not exhibit antibiotic resistance (line 29) - pathogens are resistant! b) Line 45: carbapenemase-lactamases - there are beta-lactamases and there are carbapenemases; all carbapenemases are beta-lactamases.  c) Line 46: Some class A, B and D beta-lactamases have hydrolytic activity against carbapenems. d) Line 48: "carbapenemase from K. pneumoniae": authors probably are refering to KPC (Klebsiella pneumoniae carbapenemase), which is not the only class A carbapenemase. e) authors use the term "carbapenemase genes", it should be replaced by "carbapenemase-encoding genes" f) authors are confusing susceptibility with effectiveness (line 160). g) What does "cephalosporin is inhibited by β-lactamase inhibitors" (line 192) mean? h) authors confuse existance of carbapenemase-enconding genes with their presence. 

- Line 38: Enterobacteriaceae should be written in italic letter

- Typing/spelling errors: Table 1. "Cetazidime"; it should be blaNDM, instead of blaNDM.

- Several abbreviations are not spelled out CVP, CSF (line 71, 72)

- AST (results): authors state the 100% of strains were resistant to meropenem. However, CRKP in this study are defined as resistance to meropenem; ertapenem and fosfomycin susceptibility testing should be included; susceptibility testing for newer agents suchs as ceftazidime/avibactam, meropenem/vaborbactam and/or imipenem/relebactam would be interesting; 

- bla-genes should better be called "carbapenemase-encoding genes" instead of "carbapenemase-resistance genes" since not all carbapenemases cause always carbapenem-resistance and there are other non-enzymatic mechanisms which can cause resistance to carbapenems

- intensive care units should be characterized: number of beds, surgical/trauma patients, internal medicine patients, hematopoetic stem cell transplants, solid organ transplants, etc. 

- AST (methods): Authors should state in the text that the CLSI 2019 guidelines were applied; there are no CLSI breakpoints for tigecycline, broth microdilution is not a reference method for colistin (should be stated in limitations at least);

- The authors are evaluating and comparing dianostic methods, which is beyond the scope of the proposed objectives. , e.g., AST methods such as automated BMD (by Vitek 2) and E-Test. Although the kappa-agreement is rather low (0,605) , authors state that both tests are correlating to a great degree. 

- Which MALDI-TOF equipment was used? 

Reviewer 2 Report

Brief Summary

Gandor et al. embark on a journey to measure the current antibiotic resistance in clinical isolates at the Zagazig University Hospitals. Specifically, they measure the carbapenem resistance in Klebsiella pneumonia. Gandor et al. discuss the new data compares to data from other hospitals in Egypt today and in the past.  

Significance

Emerging antibiotic resistance in clinical isolates pose a threat to modern medicine. Specifically, the rise of carbapenem resistance in Klebsiella pneumonia threatens to eliminate the usability of carbapenems as a last resort antibiotic treatment for resistant strains, especially in the intensive care units. Studies that provide an accurate snapshot of the current antibiotic resistance in hospitals is urgently needed. This type of data will be used both for informing antibiotic prescriptions and regimes in the present, as well as to develop plans to control and minimize emergence of resistance to antibiotics in the future. 

Recommendations: 

I enthusiastically recommend accepting this paper with minor revisions for publication at the Antibiotics Journal. I am listing below minor suggestions for clarifying details described in this review. I am not recommending any additional experiments.  

Notes on the text:

General: 

-       Please add a patients’ table. 

-       Please add the antibiotic type, at fist mention. 

-       Please add resistant gene mechanism. 

-       Please notice the use of “gene expression” while describing PCR results of DNA amplification of bacterial DNA, instead of “gene presence”, in the genotypic characterization part of the paper. To use gene expression there, please supplement with RNAseq, reverse transcription coupled with PCR, or MS data, which I did not see in the methods section. Please elaborate in the text, or alternatively, I suggest to adjust the text to “gene presence” in that section. 

Line 71 or Line 265: “CVP”. Please add full acronym. 

Line 74: “CRKP isolates were 100% resistant to piperacillin, piperacillin/tazobactam, cefepime, ceftazidime, aztreonam, ticarcillin, imipenem, and meropenem”, Please notice and fix the inconsistences between line 74 and table 1. Why is azithromycin missing from this list? Why is aztreonam not in the table 1? 

Line 127: “In recent years, CRKP has spread to several countries across the globe; Egypt is one of them”. Please add references that describe CRKP rise around the world.

Line 148: “Similarly, research in Indonesia and Egypt found that sputum included the highest number of bacteria with carbapenemase-encoding genes.” Please add reference. 

Line 172:” Sixty-one of the 119 isolates tested were negative for MHT. In contrast, multiplex PCR screening of carbapenemase genes revealed the presence of carbapenemase genes in 57 of the screened samples.” I read this text as 61 isolate were negative for MHT, and 58 were positive for MHT. PCR showed 57 isolates contained carbapenemase genes. 57 and 58 isolated seems very similar to me. Unless, did you mean: “ We tested the 61 MHT negative isolates by multiplex PCR screening, which revealed that 57 isolates contained carbapenemase genes. “? If yes, please clarify in your text. If no, please change “in contrast” to “high correlation with… “ or “results were consistent with …”

Line 236-241: “Nevertheless, poor infection prevention and control (IPC) efforts in developing countries have contributed to a rise in AMR prevalence [42].” To complement the paragraph on developing countries, could you please elaborate on the contribution of developed countries to the rise in AMR prevalence worldwide?

Line 246: “from various ICUs of ZUHs”. Can you please elaborate which type of ICUs, such as neonatal ICU? Or are these different locations of ICU across the country? 

Line 270:” MALDI-TOF MS were used to identify isolates”. Where are these data and results? 

Line 320: “the three significant carbapenemase genes in K. pneumoniae (blaKPC, blaNDM, blaVIM, blaIMP and blaOXA-48)”. Please notice you say "three genes" and mention 5 gene names.

No reference to table 3 in the text. 

Line 346:” MLST, PFGE, WGS”? Please use full acronyms. 

Author Response

First, thank you so much for your encourage.

Reviewer 3 Report

Dear Authors

Thank you very much for your manuscript submission. Your work is well-designed and interesting; however the presentation of your work is poor. Hence, a Major Revision is needed as below:

1. In Methodology section; Please do mention the name of city and country where you have performed your study.

2. In Results section; Please do mention the results obtained from the related tests (e.g., MALDI-TOF MS) done for detecting K.pneumoniae bacterial cells.

3. You have mentioned the number of female and male patients together with their age range in your study. However, no results are shown regarding female and male patients and their age range associated with the isolated resistant strains. It is recommended to show these data within some tables and do show the statistical correlations in this regard.

4. It is recommended to show the figures relating to E-test and Gel-electrophoresis.

5. In Results section; The mentioned Percentages should be checked in Table 2.

6. In Table 1: You have rounded the result of 106/119=89.1%; however, in "2.4. Phenotypic Carbapenemase Detection" section you did not round the result of 106/119=89.0%. There are several cases like this case. Please do revise all the cases throughout the manuscript.

7. In "2.5. Detection of Carbapenem Resistance Genes" section; You have mentioned " In 51 (48.5 %) of the isolates, carbapenemase
genes were co-expressed." It seems the calculated percentage is not correct. Please do revise it.

8. 10 or more than 10 should be written in numeral form; e.g., 12 not twelve; 26 not twenty-six etc. Please do revise all the cases throughout the manuscript

9. In table 4; Please do mention the total number of the isolates as in tables 3 and 5

10. In table 5; the last item should be rounded as 51% not 50.9% (50.98%)

11. You have mentioned "Out of 106 mCIM-positive isolates, 23 were positive for blaKPC, 26 for blaNDM, 24 for blaOXA-48, 29 for two or more carbapenemase genes, and 4 for none (Table 6)." However, table 6 shows "Positive (Resistant) (n=105)". Please do revise the aforementioned cases.

12. Please do check all the tables associated with item #11

13. The Discussion and Conclusion section should be revised in accordance with comments #3 and #18

14. Where are the used primers in your study? Please do mention all the used primers in this study (within a table) and give the related references in this regard.

15. Table 1: Ceftazidime is correct

16. You have mentioned "Coughing, dyspnea, and fever were all symptoms of pneumonia, as well as the development of an infiltrate on chest radiography and the presence of more than 104 colony-forming units per ml of purulent tracheal secretions or bronchoalveolar lavage fluid." 104 colony-forming units/ml is correct.

17. You have mentioned "Bacterial growth of more than 15 colony-forming units in roll-plate culture or 103 colony-forming units in quantitative sonication was used to identify catheter tip infection." Please do revise the related items according to comment #16.

18. Please do read and add the following papers to References section of the manuscript to have fruitful Introduction and Discussion sections:

Carbapenemase Production and Epidemiological Characteristics of Carbapenem-Resistant Klebsiella pneumoniae in Western Chongqing, China. Front Cell Infect Microbiol. 2022 Jan 4;11:775740. doi: 10.3389/fcimb.2021.775740. PMID: 35071036; PMCID: PMC8769044.

Epidemiological, Clinical and Microbiological Characteristics of Patients with Bloodstream Infections Due to Carbapenem-Resistant K. Pneumoniae in Southern Italy: A Multicentre Study. Antibiotics (Basel). 2022 May 8;11(5):633. doi: 10.3390/antibiotics11050633. PMID: 35625277; PMCID: PMC9137758.

Virulence factors, antibiotic resistance patterns, and molecular types of clinical isolates of Klebsiella Pneumoniae. Expert Rev Anti Infect Ther. 2022 Mar;20(3):463-472. doi: 10.1080/14787210.2022.1990040. Epub 2021 Oct 28. PMID: 34612762.

The Rapid Emergence of Ceftazidime-Avibactam Resistance Mediated by KPC Variants in Carbapenem-Resistant Klebsiella pneumoniae in Zhejiang Province, China. Antibiotics (Basel). 2022 May 30;11(6):731. doi: 10.3390/antibiotics11060731. PMID: 35740138; PMCID: PMC9219983.

Aminoglycoside-resistance gene signatures are predictive of aminoglycoside MICs for carbapenem-resistant Klebsiella pneumoniae. J Antimicrob Chemother. 2022 Feb 2;77(2):356-363. doi: 10.1093/jac/dkab381. PMID: 34668007; PMCID: PMC9097246.

Metallo-ß-lactamases: a review. Mol Biol Rep. 2020 Aug;47(8):6281-6294. doi: 10.1007/s11033-020-05651-9. Epub 2020 Jul 11. PMID: 32654052.

Liu C, Dong N, Chan EW, Chen S, Zhang R. Molecular epidemiology of carbapenem-resistant Klebsiella pneumoniae in China, 2016–20. The Lancet Infectious Diseases. 2022 Feb 1;22(2):167-8.

Kalu M, Tan K, Algorri M, Jorth P, Wong-Beringer A. In-Human Multiyear Evolution of Carbapenem-Resistant Klebsiella pneumoniae Causing Chronic Colonization and Intermittent Urinary Tract Infections: A Case Study. Msphere. 2022 May 9:e00190-22.  

Prevalence of Tetracycline Resistance Genes tet (A, B, C, 39) in Klebsiella pneumoniae Isolated from Tehran, Iran. Iranian Journal of Medical Microbiology. 2022 Feb 10;16(2):141-7.  

Cost-utility analysis of ceftazidime-avibactam versus colistin-meropenem in the treatment of infections due to Carbapenem-resistant Klebsiella pneumoniae in Colombia. Expert Rev Pharmacoecon Outcomes Res. 2022 Mar;22(2):235-240. doi: 10.1080/14737167.2021.1964960. Epub 2021 Aug 18. PMID: 34407710.

Round 2

Reviewer 1 Report

1.       Language quality has to be improved, see L 17, L19, L29, L 128, and others

2.       The introduction and discussion are too long and not really straight forward. E.g., authors explain Ambler Classification twice!

It is unnecessary to explain β-lactamases in detail and to name different carbapenemases. Instead, I would suggest focusing on the problem of carbapenem resistance, briefly review mechanisms of carbapenem resistance (in one or two sentences) and “the big five” carbapenemases.

Again, although the results are concerning, in my opinion this study is lacking novelty and is only a single center study which is not evaluating newer therapeutic options such as ceftazidime/avibactam, cefiderocol, etc. 

Furthermore, authors performed and compared several phenotypic test which are unncessary and have been evaluated and discussed extensively. 

I would suggest to publish this research as a short report, focusing only on antimicrobial susceptibility testing and results of PCR. 

There are still several other shortcoming. To name a few:

L 16: a) Don´t use abbreviations if they are not used again in the abstract “(ZUHs)”; b) authors still use the term “existence” of carbapenemase-encoding genes instead of “presence”

L 17: a) Start sentences with capital letters; b) Carbapenemase-encoding genes were “found/ were present” instead of “presented”; c) I would suggest not using the term “significant” since it is often used in the context of statistics, such as statistically significant, instead I would suggest using “in a high number” or similar expressions.

L 18: “All isolates were resistant to commonly used antibiotics”: It is true, but since this study is evaluating only CRKPs, this statement is unnecessary.

L 19: “this condition”: I would suggest finding another word, since condition is often used as a synonym of illness

L 25: Eliminate “(K. pneumoniae)”.

L 29:  a) It is one of the most [common/frequent/…] pathogens; b) exhibits resistance to multiple antibiotics

L 31: a) “Antibiotics resistance genes”; I would suggest using “genetic resistance determinates” since not all genetic resistance determinants lead to phenotypical resistance, while physiological changes or production of biofilms can also lead to antimicrobial resistance…

L 231: blaKPC is a carbapenemase-encoding gene, not a carbapenemase!

Correction: Althoug there was some discussion about colistin susceptibility testing and the interference of plastic, BMD is the reference method for colistin susceptibility testing!

Reviewer 3 Report

Dear Authors

Thank you for your effective revision. My decision regarding your revised manuscript is "Accept"; however, I suggest you to read and add the following papers to References sections to have fruitful Introduction and Discussion sections.

Virulence factors, antibiotic resistance patterns, and molecular types of clinical isolates of Klebsiella Pneumoniae. Expert Rev Anti Infect Ther. 2022 Mar;20(3):463-472. doi: 10.1080/14787210.2022.1990040. Epub 2021 Oct 28. PMID: 34612762.

Metallo-ß-lactamases: a review. Mol Biol Rep. 2020 Aug;47(8):6281-6294. doi: 10.1007/s11033-020-05651-9. Epub 2020 Jul 11. PMID: 32654052.

Author Response

Thanks you very much for your encouraging reply, and thanks again for your valuable suggestions which were rich in useful information.

We added from it to the discussion with  a reference 50. 

Round 3

Reviewer 1 Report

In my opinion the manuscript has improved a lot. However, it still contains several errors and misconceptions. In my opinion the manuscript should not be accepted in the current form. Nevertheless, I would like to encourage the authors to re-write the manuscript and re-submit the alarming results as a short report.

L 41: AmpC β-lactamases

L 72: β-lactam

L 75 ff: I’d suggest something shorter, like: “Carbapenemases can be classified in three functional classes: Class A…”; I’d rather mention only the most relevant carbapenemases like the “big five”, which are evaluated in the study.

L: 92: Why “nevertheless”?

L 152: “According to the meropenem (MEM) E-test results, MEM was…”. Please do not use terms like “effective” or “ineffective”. I’d rather suggest some like: “110/180 were meropenem resistant” (or you could also use the term “non-susceptible” if you include intermediate isolates. Instead of the MIC ranges it might probably more useful to state MIC50.

L 167: “harbored”; “evaluated carbapenemase-encoding genes”

L 171: “co-presented” sounds strange. I’d suggest something like “22/105 isolates co-harbored…”

L 173: presented both (deleted “the”) blaKPC and blaOXA-48.

L 512: Class A carbapenemases are NOT inhibited efficiently by clavulanic acid or tazobactam”

L 522: Again, DO NOT USE the term existance

L 536: You probably mean “unlike in other studies”; unlikely means “not likely to happen, be done, or be true; improbable”.

L543: Bacteria either express NDM or harbor blaNDM but they do not “generate” genes

L 819: As mentioned in my last review, BMD is a method for colistin resistance; therefore, using Vitek 2 for AST is valid and does not need to be mentioned in limitations.

Suggestion: Authors should also provide a clean version of the manuscript. 
